# Assisted Robust Reward Design

**Jerry Zhi-Yang He**
Electrical Engineering and Computer Science
University of California Berkeley
hzyjerry@berkeley.edu

**Anca D. Dragan**
Electrical Engineering and Computer Science
University of California, Berkeley
anca@berkeley.edu

**Abstract:** Real-world robotic tasks require complex reward functions. When we define the problem the robot needs to solve, we pretend that a designer specifies this complex reward exactly, and it is set in stone from then on. In practice, however, reward design is an *iterative* process: the designer chooses a reward, eventually encounters an "edge-case" environment where the reward incentivizes the wrong behavior, revises the reward, and repeats. What would it mean to rethink robotics problems to formally account for this iterative nature of reward design? We propose that the robot not take the specified reward for granted, but rather have *uncertainty* about it, and account for the future design iterations as *future evidence*. We contribute an Assisted Reward Design method that speeds up the design process by anticipating and *influencing* this future evidence: rather than letting the designer eventually encounter failure cases and revise the reward then, the method actively exposes the designer to such environments during the development phase. We test this method in a simplified autonomous driving task and find that it more quickly improves the car's behavior in held-out environments by proposing environments that are "edge cases" for the current reward.

**Keywords:** Reward Design, Safety, Human-in-the-loop

## 1 Introduction

One popular way to design robotic agents is to program them to maximize the cumulative reward in their environments. While we have made great progress in solving this optimization problem, we often delegate finding a good reward function to the human designer. In reality, reward design is an iterative and challenging process. Human designers spend tremendous effort iterating on the reward function as they test the robot in more and more environments. What is more, once deployed, reward functions can still be incorrect in edge-case environments, leading to failures.

Take, for instance, autonomous cars. They have to correctly balance safety, efficiency, comfort, and abiding by the law. A human designer starts by looking at some representative traffic scenarios and specifying a reward function that leads to desired behaviors in each of them. But working well in this set of environments is not enough — the reward function has to incentivize the right behavior in *any* environment the car will encounter in its lifetime[1]. The engineer will test-drive the car in both simulation and in the real world. Almost inevitably, optimizing for the specified reward will lead to some undesirable behavior in some further environments. The engineer will then *revise* the initial reward and repeat the process. If they are lucky, they will have converged to a suitable reward function by deployment time. But for many systems, this is a never ending process: the car will encounter some edge-cases in the real world that they did not foresee — an unseen road layout or traffic situation — and the engineer revises the reward again on the new environment.

In this work, we explore what it would mean for the robot to explicitly account for this iterative nature of reward design. Rather than treating the current reward as set in stone, we envision a system in which the robot works together with the designer and helps them design the right reward function, as in Fig. 1: the robot takes the reward function specified by the designer, and proposes a new environment for the designer to investigate where the reward function might fail. To capture this formally, we first enable the robot to explicitly account for the *uncertainty* in the reward function

---

[1]While there are challenges to building a world model, planning algorithms, etc., for this paper, we assume success in that and focus on designing the reward function.

5th Conference on Robot Learning (CoRL 2021), London, UK.

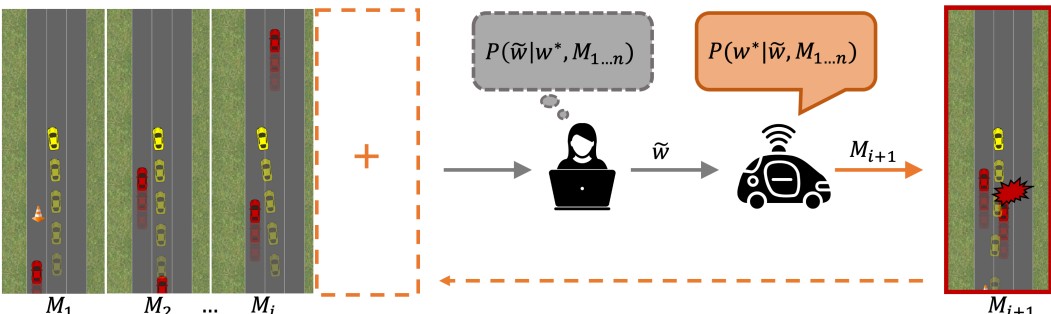

Figure 1: In the Assisted Reward Design process, the designer specifies a proxy reward $\tilde{w}$ on a set of environments $M_{1,...n}$. The robot agent takes the current proxy, infers about the true reward $w^*$ and queries the designer with a new environment where the proxy fails. The designer then revises her proxy to work on this new environment.

itself: every reward revision does *not* define the reward function fully, but is instead an *evidence* about it. Previous work on Inverse Reward Design [1] incorporates this uncertainty into the reward function with the goal of making the robot plan more conservatively in new environments. In this work, rather than passively updating the robot's estimate of the reward, we argue that it is the robot's job to actively help the engineer design more robust reward functions, with the goal of performing well at deployment time. We refer to this as *Assisted Reward Design*. The key insight is that:

*By accounting for the iterative nature of reward design, the robot does not just passively estimate the reward it receives but proactively influences the future evidence it can gather from the designer.*

Our contributions are three-fold:

**1. Formalizing reward design as an iterative process.** We formalize the problem of iterative reward design as a Meta MDP with hidden state. Under this formulation, the robot can and should account for the future iterations of the reward as *future evidence*. The robot can then make these pieces of evidence more informative by exposing the designer to environments where revisions are most likely needed. Rather than letting the designer eventually encounter edge-case environments, our method *proposes* candidate environments and asks for reward designs, inserting itself in the design process loop, as in Fig. 1. This has two implications. In cases where the designer would converge to a good reward before deployment time, our method speeds up the iterations by focusing on informative environments. Perhaps more importantly, in cases where the designer might still otherwise have the wrong reward when deploying the system, our method has the potential to expose the edge-case environments that cause failures, prompting fixes in the reward ahead of time.

**2. Providing an approximate algorithm for our method.** It is intractable to compute the reward uncertainty given the high-dimensional nature of the reward space and the environment space. We propose an approximate algorithm based on particle filtering to enable efficient computation.

**3. Analyzing Assisted Reward Design in a simplified autonomous car domain.** We evaluate our method's practicality and effectiveness in designing reward functions for autonomous cars, using the open-source environment proposed by Sadigh et al. [2]. We conduct experiments with both simulated reward designers and robotics experts. We find that our method uncovers more difficult test scenarios and speeds up the reward design process. We also showcase that it proposes interesting environments that tend to be *edge cases* where the current reward estimate fails.

Overall, this paper takes a step towards making our robotic systems understand reward specification as an iterative process. In the end, we would like robotic systems not just to optimize what we design for them but help us design better and more robust reward functions. Even though admittedly more work is needed to put these ideas in real-world systems, especially in light of the need for a simulator or learned model of environments, we are excited to see our method assist in difficult reward design tasks and produce *edge-case environments* without any hand-coded heuristics.

## 2   Related Work

Designing robots to work reliably in an open world remains a challenging problem. Our work bridges three lines of work towards this goal: teaching robots the correct behaviors, enabling robots to actively learn from humans, and ensuring safety and robustness when deploying robots.

**Learning Rewards from Diverse Human Input** Given the difficulty of specifying reward functions, researchers have explored a wide variety of different methods to enable robots to learn rewards instead

from human input, especially demonstrations [3, 4, 5, 6]. Using this idea, recent works have explored learning from a diverse set of modalities. These include language instructions [7, 8], numerical feedback [9, 10], comparisons [11, 12], advice [13], facial expressions [14], collaborative behavior [15], or corrections [16, 17]. In contrast, our work builds on [1], which uses the specified reward function itself as the human input to learn from. We generalize this idea to an iterative, active process.

**Active Reward Learning** Our work is an instance of active reward learning. Prior work has focused on actively querying for comparisons [18, 19], demonstrations [20, 21, 22, 23], or reward labels [24]. Such queries are typically for trajectories or states in a given environment (MDP). Recent work has proposed active learning to synthesizes the environment itself to gain more information about the reward (either the initial configuration and behavior of other agents in the scene, such as [12], or the entire environment [24]). We also propose to actively query new environments, but the human feedback is a proxy reward designed to work only in that environment. This is interesting for two reasons. First, we show that it leads to the robot proposing edge cases to the designer where the previous reward is likely to fail, thus aiding in revising the reward. Second, we argue that the same active reward learning ideas from the literature can be applied to the reward engineering process — when we design a reward function, we should not just have the robot go optimize it blindly. We should instead think about it as an iterative process through the lens of active reward learning.

**Reward Shaping** Many prior works have also focused on recovering reward functions for suboptimal planners, given a ground truth reward function that is difficult to optimize, e.g. due to sparsity or non-differentiability [25, 26]. Note that in many real-world problems, such as autonomous cars, the ground truth reward is not obvious to write down. Assisted Reward Design is different from the reward shaping problem in that we do not assume access to a ground-truth reward function, but seek techniques that leverage proxy rewards that are the designer's best guesses.

**Safety in Real-World Robots** Robots deployed in the open-world need to handle a large set of possible environment configurations, be it a busy road for an autonomous car or a messy kitchen for a home robot. Given this challenge, there has been a rising number of work on safeguarding robot behaviors by generating edge cases. Koren et al. [27] and O'Kelly et al. [28] studied using black-box optimizers to perform adaptive testing and generate failure cases. [29] trains a generative model of environments from datasets. Uesato et al. [30] generates edge cases by learning a failure predictor. These methods require defining hard constraints that the robot should not violate. Similarly, reward shaping work has looked at how to incorporate such constraints into the reward [31].

In contrast, we focus on settings where such hard constraints are incomplete or difficult to specify. Our method does not assume access to such constraints. We show that our method can learn to develop edge-case scenarios that implicitly violate constraints without having to specify them.

## 3 Problem Statement

We first describe the Unassisted Reward Design problem works, then we formulate the Assisted Reward Design problem and introduce our method.

### 3.1 The Unassisted Reward Design Problem

**Problem Setup** Let an "environment" $M$ be a Markov Decision Process without the reward function. At development time, we assume access to a large set of such environments, $\mathcal{M}_{\text{devel}}$. We assume this is a large set that the designer cannot exhaustively test. This can be, for instance, all the environments in a several-million-mile autonomous driving dataset. Or, it can be a parameteric set of possible environment configurations, such as placements of objects, roads, and other agents. Finally, it can also be a generative model of the world. The robot will run at deployment time in a different set of environments $\mathcal{M}_{\text{deploy}}$, which we do *not* have access to during development time. [2]

We further assume access to a space of reward functions parameterized by $w \in W$ — these can be linear weights on pre-defined features or weights in a neural network mapping raw input to scalar reward. We denote by $w^*$ weights of the ground truth reward function, which induces the desired behavior in $\mathcal{M}_{\text{devel}}$ and $\mathcal{M}_{\text{deploy}}$. We denote by $\xi_{w,M}$ the trajectory (or policy, but for this work, we consider deterministic MDPs with set initial states) optimal to the cumulative reward induced by $w$

---

[2]Our method works best when we can induce a vast set $\mathcal{M}_{\text{devel}}$ that includes all possible scenarios in $\mathcal{M}_{\text{deploy}}$. Otherwise, if $\mathcal{M}_{\text{deploy}}$ is allowed to be drastically different, and if $\mathcal{M}_{\text{devel}}$ is not enough to design a robust reward function, this exposes the robot to unanticipated failures after deployment.

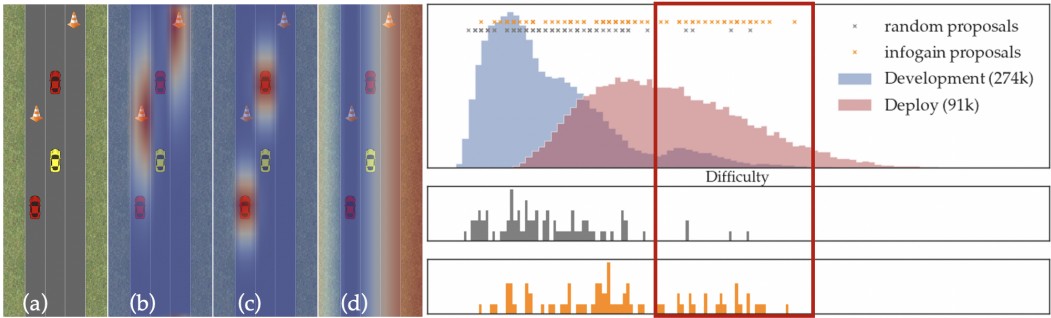

Figure 2: Left: (a) The driving environment with autonomous vehicle (yellow) and human-driven vehicles (red). (b)(c)(d) highlights selected features: obstacle distance, vehicle distance, progress. Right: the distribution of $\mathcal{M}_{\text{devel}}$ (blue) and $\mathcal{M}_{\text{deploy}}$ (pink) based on difficulty metric.

in $M$, $\xi_{w,M} = \arg\max_\xi R_w(\xi; M)$. Our assumption is that there exists a $w^*$ such that $\xi_{w^*,M}$ is the desired behavior for any $M \in \mathcal{M}_{\text{devel}}$ and any $M \in \mathcal{M}_{\text{deploy}}$. We do *not* have access to $w^*$.

**The Process of Unassisted Reward Design** The designer takes a subset of environments $\mathcal{M} \subseteq \mathcal{M}_{\text{devel}}$ and specify a reward function (via parameters $\tilde{w}$) that leads to good behavior in those environments. In a one-shot design, they would select a set $\mathcal{M}$, assume it is representative of $\mathcal{M}_{\text{deploy}}$, design a $\tilde{w}$ for that $\mathcal{M}$, and deploy the system with $\tilde{w}$ (never changing the reward again). However, in reality, this is an iterative process. They start with an $\mathcal{M}_0$, design a $\tilde{w}_0$, and eventually encounter during their testing a new environment $M'$ on which optimizing $\tilde{w}_0$ does not lead to desirable behavior. Then, they would augment their set to $\mathcal{M}_1 = \mathcal{M}_0 \cup \{M'\}$, and re-design the reward: $\mathcal{M}_0 \to \tilde{w}_0 \to \mathcal{M}_1 \to \tilde{w}_1 \to \dots$. This might continue into deployment if the reward at the time of deployment still fails on $\mathcal{M}_{\text{deploy}}$. Implicitly, at every step along the way, the robot treats the current $\tilde{w}_i$ as equivalent to $w^*$.

### 3.2 The Assisted Reward Design Problem

We formulate the Assisted Reward Design Problem by introducing a meta-MDP problem, where the meta-agent receives proxy rewards as observations from the designer and acts by proposing new environments. The goal is to minimize regret of the reward design over deployment $\mathcal{M}_{\text{deploy}}$.

**Meta-States, Meta-Actions, Observations, Transitions.**[3] In the Assisted Reward Design problem, the notion of a "state" is a set $\mathcal{M}$ of environments — this is what changes over time and is directly observable — along with the hidden state $w^*$. The meta-agent starts at state $(\mathcal{M}_0, w^*)$ (environments either chosen by the designer to be representative, or simply the empty set). It transitions to the next state by "acting", i.e. adding one more environment $M_i$ to the set: $\mathcal{M}_{i+1} = \mathcal{M}_i \cup M_i$. Every time it enters a state $(\mathcal{M}_{i+1}, w^*)$, it receives an observation $\tilde{w}_{i+1}$ about $w^*$ from the designer. For the meta-agent to assist the designer, it needs to treat the observed reward $\tilde{w}_i$ not the same as $w^*$, but as a proxy that emulates $w^*$ *only* on the current $\mathcal{M}_i$. It can then interpret $\tilde{w}$ as evidence about $w^*$ and use all past evidence to obtain a belief of $w^*$. To assist the designer, the key step is to account for (and thus influence) the future evidence in order to get a better reward estimate.

**Observation Model.** Reward designers tend to select proxy rewards that work well in training environments. With this intuition, previous work [1] introduced an observation model that computes the probability of observing proxy reward $\tilde{w}$ under $w^*$ and designer's training environment set $\mathcal{M}$:

$$P_{\text{design}}(\tilde{w}|w^*, \mathcal{M}) \propto \exp\left[\sum_{i=1}^{|\mathcal{M}|} \beta R_{w^*}(\xi_{\tilde{w}_i, M_i})\right] \tag{1}$$

with $\beta$ controlling how close to optimal we assume the person to be. This distribution informs the meta-agent that it is receiving proxy rewards from an approximately optimal designer who looks at the current $\mathcal{M}$. We can then leverage this model to infer a belief over $w^*$: $b_\mathcal{M}(w^*)$.

**Objective.** The objective of Assisted Reward Design is to achieve high reward at deployment time. What the meta-agent controls are a sequence of actions, i.e. a sequence of environments $\mathcal{M}_T$ it can propose. These lead to observations about $w^*$. At deployment time, the meta-agent uses its belief to generate optimal trajectories and accumulates reward according to $w^*$. We want to find the set of environments $\mathcal{M}_T$ such that when given to the designer, these environments induce a belief that leads to the best performance in $\mathcal{M}_{\text{deploy}}$:

$$\max_{\mathcal{M}_T} \mathbb{E}_{M \sim \mathcal{M}_{\text{deploy}}} \mathbb{E}_{w \sim b_{\mathcal{M}_T}(w^*)} R_{w^*}(\xi_{w,M}) \tag{2}$$

---

[3]For the rest of this paper, all our references to "state" and "action" stand for "meta state" and "meta action".

where $b_{\mathcal{M}_T}(w^*)$ is the meta-agent's belief at deployment time, based on the evidence gathered from the states $\mathcal{M}_0, .., \mathcal{M}_T$; $\xi_{w,M} = \arg\max_{\xi} R_w(\xi; M)$ is the optimal trajectory.

As currently stated, this objective is impossible to optimize directly because the meta-agent does not know $\mathcal{M}_{\text{deploy}}$. In what follows, we introduce a solution to Assisted Reward Design based on the heuristic objective of identifying $w^*$ — if the meta-agent uses its actions to disambiguate what $w^*$ should be, then it can achieve optimum performance on the objective in Eq. 2 because it can plan optimally to $w^*$. We first introduce the Maximal Information objective. Then we show how the meta-agent can track a belief and perform future belief updates in order to propose new environments.

## 4   Assisted Reward Design via Active Info-Gathering.

**Maximal Information.**   We invert the probability model of Eq. 1 to obtain a belief over $w^*$: $P(w^*|\tilde{w}, \mathcal{M})$. Our key insight is that we can reduce the belief uncertainty by proposing future environments to influence the designer. In other words, we would like to find environments that can disambiguate the reward as much as possible — if we can identify $w^*$ exactly, the meta-agent attains zero regret at deployment. Thus, we introduce an approximate strategy for the meta-agent: proposing environments that provide the most information gain over its belief distribution, as commonly done in active learning [32, 33] and robot active exploration [34]. For computational tractability, we do a one-step look-ahead only, although the meta-agent could benefit from reasoning over multiple future iterations. Given an action $M$ (MDP), we compute its mutual information with $w$:

$$f(M_{i+1}) = \mathbb{I}[w; M_{i+1}] = \mathbb{H}[w|\mathcal{M}_{0:i}, \tilde{w}_{0:i}] - \tag{3}$$
$$\mathbb{E}_{w^* \sim P_i(w), \tilde{w}_{i+1} \sim P_{\text{design}}(\tilde{w}_{i+1}|w^*, M_{i+1})}[\mathbb{H}[w|\mathcal{M}_{0:i+1}, \tilde{w}_{0:i+1}]]$$

Here $\mathbb{H}[w|\mathcal{M}_{0:i}, \tilde{w}_{0:i}]$ is the entropy of posterior distribution of $P_i(w)$, and $\tilde{w}_{i+1}$ is a new observation sampled from the updated designer model $P_{\text{design}}(\tilde{w}_{i+1}|w^*, M_{i+1})$. After observing $\tilde{w}_{i+1}$, we can compute the conditional entropy $\mathbb{H}[w|\mathcal{M}_{0:i+1}, \tilde{w}_{0:i+1}]$ using methods from Sec. 4.1. In our implementation, we set $\tilde{w}_{i+1} = w^*$ to make Eq. 3 more tractable and find that it empirically works well. Next we introduce two update rules to compute conditional entropy — joint and independent mode.

### 4.1   Updating Beliefs over $w^*$

**Joint reward design.**   At iteration $i+1$ of Assisted Reward Design, the meta-agent takes an action $M_{i+1}$ (proposes an environment), and receives an observation $\tilde{w}_{i+1}$ – a reward that works over *all* environments $\mathcal{M}_{i+1} = \mathcal{M}_i \cup \{M_{i+1}\}$. Assuming that $\tilde{w}_{i+1}$ is independent from previous observations, we can use $\tilde{w}_{i+1}$ to update the belief over $w$ using Eq. 1:

$$P_{i+1}(w|\tilde{w}_i, \mathcal{M}_i) \propto P_i(w)P_{\text{design}}(\tilde{w}_i|w, \mathcal{M}_i) \tag{4}$$

with $P_0$ the meta-agent's prior (uniform in our experiments). We take a sequential-importance-resampling particle filtering approach to approximate $P_{i+1}$: we sample $N_p$ particles from $P_i$, compute importance weights based on Eq. 1 for each particle, and resample using the importance weights. In order to compute the normalized $P_{\text{design}}$, we approximate the normalizer in Eq. 1 via samples.

**Independent reward design.**   We also allow designers to provide an observation $\tilde{w}_i$ based solely on the new environment $M_i$. Prior work [35] finds that this model puts less burden on the designer. Under this assumption, our belief update simplifies to relying on a single new environment:

$$P_{i+1}(w|\tilde{w}_i, M_i) \propto P_i(w)P_{\text{design}}(\tilde{w}_i|w, M_i) \tag{5}$$

### 4.2   Adding New Features

Reward engineering often requires adding new features to the environment. Our method is compatible with the designer adding features. See appendix for more details.

## 5   Experiments with Simulated Designers

We evaluate Assisted Reward Design in a simplified autonomous driving task. In this section, we hypothesize a ground truth reward function and simulate designer behaviors. Note that our method does not require such unique ground truth $w^*$ – this is merely used here for evaluation. In Sec. 6, we conduct an expert study where we do not have the ground truth.

### 5.1   Experiment Setup

**Driving task.**   As illustrated in Fig. 2, the autonomous car is to merge to the leftmost lane. Since there may be other human vehicles (simplified as constant speed) and obstacles, the autonomous car

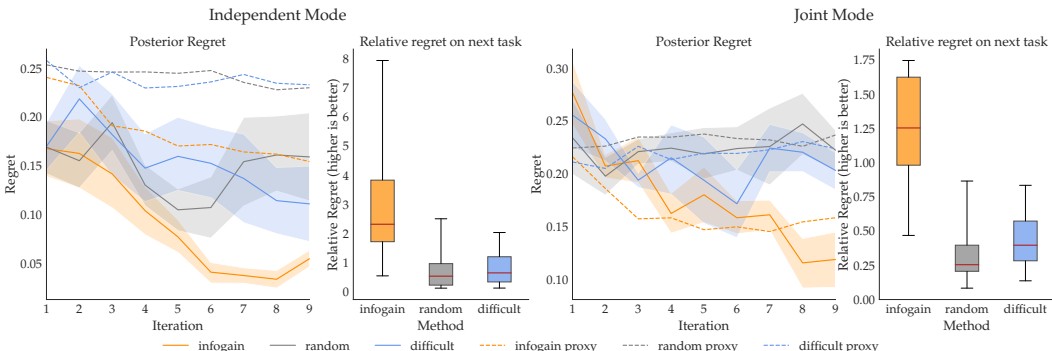

Figure 3: Experiment with simulated designers. Left: independent mode, right: joint mode. (a)(c) shows the regret of the posterior mean compared to true reward under different acquisition functions. (b)(d) shows the efficacy of the proposed environment. The "box" denotes 5th to 95th percentile.

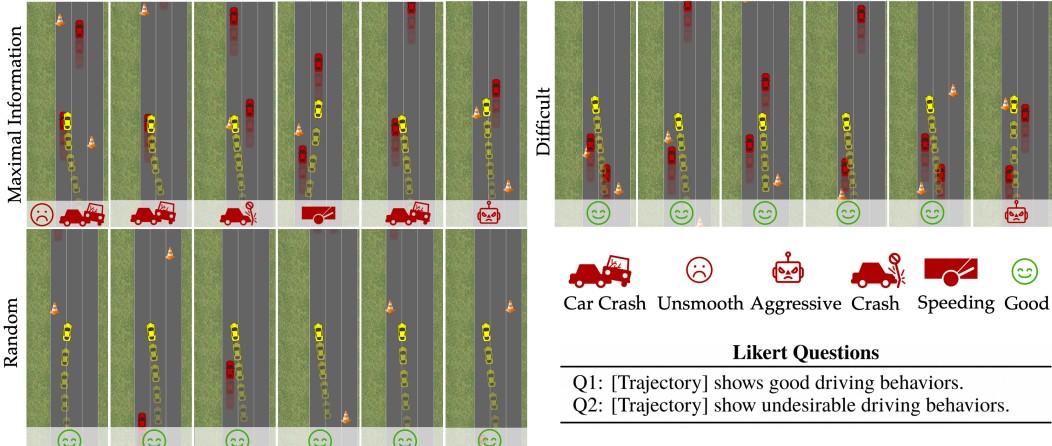

Figure 4: Visualization of the top proposed environments based on each acquisition function.

needs to decide whether to overtake or to slow down. Our reward function based on Sadigh et al. [2] is a weighted sum of six features, including vehicle distance, progress, control effort, etc. Please refer to supplementary material for their details. A good reward function should correctly balance these terms and reflect our desired behaviors in all different situations. To obtain $\mathcal{M}_{\text{devel}}$, we parametrize environments over the initial placements of obstacles and other vehicles.

**Environment Difficulty.** Successful merging tends to be more difficult when human vehicles and obstacles are closer to the autonomous vehicle. We can use a simple metric to describe each environment: difficulty $= \sum_i 1/d_{\text{human vehicle}_i} + \sum_j 1/d_{\text{traffic cone}_j}$. We visualize the distribution of $\mathcal{M}_{\text{devel}}$ and $\mathcal{M}_{\text{deploy}}$ in Fig. 2 using this metric. Note that both $\mathcal{M}_{\text{devel}}$ and $\mathcal{M}_{\text{deploy}}$ have a "long-tail" distribution of events — events that are rare to encounter daily, but could be highly useful for the reward design. An ideal acquisition function in should be able to identify these events.

**Baselines.** We compare our Maximal Information acquisition function with two baselines that represent common design paradigms where the designer selects (1) the most difficult tasks ranked by $M \in \mathcal{M}_{\text{devel}}$ (2) random environments from $\mathcal{M}_{\text{devel}}$.

## 5.2 Experiment Results

**Evaluation.** A good Assisted Reward Design method should help the reward designer quickly recover high-quality reward functions. To evaluate this, we specify one set of ground truth reward $w^*$. At every iteration, we simulate the designer using $w^*$ under both *independent* mode and *joint* mode as described in Sec. 4.1. Please refer to the appendix for experiment details. We plot the performance of posterior mean and the proxy reward $\tilde{w}$ under different methods in Fig. 3. To mitigate the effect of the choice of initial environments, we experiment on six random initial environments and find that our active method outperforms the proxy reward $\tilde{w}$, as well as the random and difficult baseline in both joint and independent modes (p < 0.05).

Our active method (orange) outperforms both the random baseline (grey) and the difficulty heuristic (blue) as shown in Fig. 3. This means that certain environments, despite being rated "easy" by heuristics, turn out useful for the reward design. These environments are often edge-cases that subtly trigger undesirable behaviors — in Fig. 1 a rather empty road with two human vehicles triggered the autonomous vehicle to crash. Next we further investigate this.

**Edge-Case Nature of Proposed Environments** To study why our method performs better than the baselines, we measure the quality of the proposed environments using: $r(M_{\text{next}}) = \mathbb{E}_{\tilde{w} \sim P(w=w^*)}\left[\text{Regret}(\tilde{w}; w^*, M_{\text{next}})\right] / \mathbb{E}_{\tilde{w} \sim P(w=w^*)}\mathbb{E}_{M \sim \mathcal{M}_{\text{deploy}}}\left[\text{Regret}(\tilde{w}; w^*, M)\right]$, the ratio of regret on next environment versus average regret in deployment. Higher $r(M_{\text{next}})$ means the next environment better uncovers the overall regret of the current posterior. As shown in Fig. 3, our active method proposes environments of highest relative regret. This suggests that to find the most useful environment for Assisted Reward Design, simply relying on heuristic environment ranking is not enough. It is more effective to utilize all proxy rewards and their induced uncertainty over $w^*$.

# 6 Experiments on Real Reward Design

We conducted a study with 14 robotics experts[4] on the simplified autonomous driving tasks, where we ask them to design reward functions to generate their preferred driving behaviors.

## 6.1 Experiment Setup

We give the experts 10 minutes to familiarize with the features. We then let them design reward functions over four consecutive iterations, under three different methods and three different random seeds. This gives a total of 36 individual reward designs. We use counterbalancing to randomize the sequence of different methods. All our experiments are conducted over Zoom.

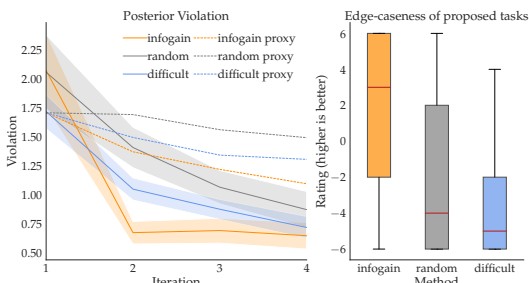

Figure 5: Experiment with robotic experts. The left shows the violation under different acquisition functions. The right compares the relative value of proposed environments.

**Violation.** Since we do not have the ground truth reward function, we cannot measure posterior regret as in Sec. 5. For evaluation purposes only, we introduce seven violation criteria to measure the quality of our reward design, including collision, driving off track, stopping, etc. Please refer to the appendix for detailed definitions. We use these criteria as unit tests on the behavior to capture unwanted behaviours. We then count the total number of violations of each trajectory. Better reward functions should induce lower violation counts. Although we could explicitly incorporate these tests as hard constraints in the "reward" function [31], we keep the reward features oblivious to them. This is for two main reasons. First, in the real world, constraint violations are acceptable in some edge-case scenarios: slamming on the brake, despite uncomfortable, is justifiable at the sight of traffic accidents. Such trade-offs are difficult to specify and it is our goal to exploit potential edge-cases. Second, violation tests only capture a subset of unwanted behaviors. Tailgating, for instance, does not break the tests. Yet, it is undesirable in most cases for the autonomous vehicle. In Sec. 6.2 we provide qualitative examples from the experiment to illustrate such undesirable behaviors.

## 6.2 Experiment Results

**Evaluation.** We perform Assisted Reward Design where experts answer the queries stemming from our algorithm, as well as the baselines. We visualize the result in Fig. 5 left. Maximal Information outperforms the other acquisition functions in causing fewest average violations on $\mathcal{M}_{\text{deploy}}$. This shows that Assisted Reward Design can help us tune a reward function that fails fewer tests as a measurable proxy for the good driving behaviors in the real world.

**Edge-Case Nature of Proposed Environments.** To measure the quality of the proposed environments at each iteration, we ask the experts to watch the trajectories that their proxy designs generate on the proposed tasks. We then use the Likert scale questions in Fig. 4 and ask the users to rate how much they agree with the statement on a scale of 1-7. We then compute the normalized score $Q1 - Q2$ as a measure of how good the behavior is. A lower score indicates better behavior. We use the inverse of the score to indicate edge-caseness in Fig. 5. Our maximal information method outperforms the heuristic difficulty metric in finding edge-case environments that contain more violations.

---

[4]from the authors' institution

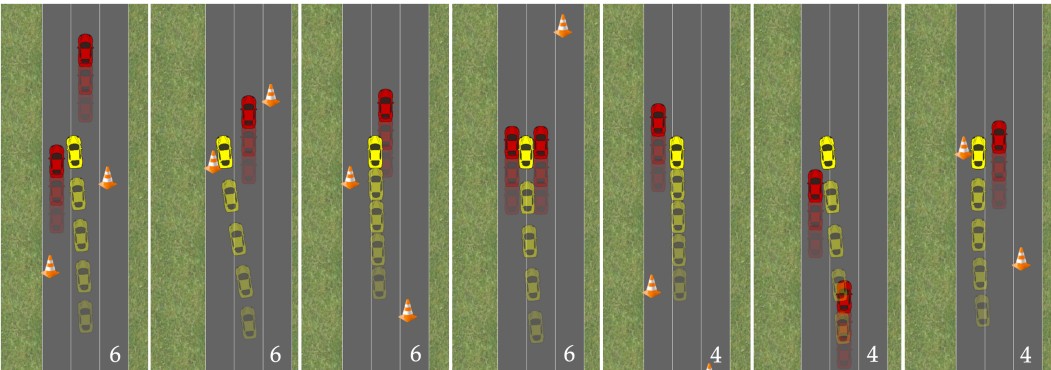

Figure 6: Selected environments proposed by Maximal Information with the highest rating by users. The robot plans a trajectory using previously designed $w_{\mathrm{MAP}}$. Normalized Likert scale rating is shown on the bottom right of each scenario. Higher rating means more likely to be an edge case. Notice that these trajectories are failure cases but do not violate the pre-defined difficulty metrics such as collision.

**Qualitative Analysis of Proposed Environments** We randomly sample an initial environment and query designer. We then use maximal information, heuristic difficulty, and random acquisition function to propose the next environments. We visualize the trajectory of MAP estimate $w_{\mathrm{MAP}}$ on each of these new environments in Sec. 6. Notice that interestingly, our method tends to find environments where the current MAP estimate fails, even though we do not explicitly optimize for finding failure cases. It turns out that these environments help reward designers effectively narrow down on the true reward. In comparison, heuristic difficulty finds environments with dense interactions. Yet, they do not induce failure in the resulting trajectories likely because such difficulties have been addressed by past reward designs. Heuristic difficulty tends to generate repetitive environments.

**Can Violation Count Replace the Maximal Information?** Sec. 6.2 uses violation counts to evaluate trajectories. One may ask the question: can we simply use the violation count as an acquisition function to find edge-case environments? In other words, environments where the design leads to high collision, off-track counts, etc. should be proposed as edge cases. On a surface level, this seems to be a reasonable approach. In fact, this is commonly used in industry [36]. However, we argue that setting violation counts as the acquisition function limits the scope that we can identify edge cases. We highlight this in Fig. 6, where we visualize selected environments proposed by Maximal Information. They exhibit subtle failures and lead to the highest Likert-scale ratings despite obeying all violation criteria. Interestingly in the second, fourth, and seventh examples, the vehicle narrowly avoids the collision. Such behaviors found by Maximal Information are hard to capture with hand-designed violation metrics. By acting on the edge of violations, the algorithm pushes the reward designer to better specify her "decision boundaries". As a result, Maximal Information can propose more diverse edge cases and gather designer preference more quickly.

## 7 Discussion

We contribute Assisted Reward Design, an active reward learning method that generates environments and queries for reward functions as user inputs. We find experimentally that our method exposes edge cases — environments where the current proxy reward leads to high regret. Experiments with robotic experts confirmed that these environments are edge cases that lead to undesirable behaviors.

Our method is applicable to model predictive control problems such as drone, indoor robot navigation, motion planning for manipulation [35] where we have known dynamics, as well as RL problems given the environment simulator or learned model. Given that naively applying RL in the inner loop of reward inference is intractable, it remains an open question how to efficiently combine with RL.

There are several limitations to our approach. Firstly, our environment proposal algorithm could be more efficient. One may explore synthesis and continuous optimization over the environment parameters. Secondly, we need of formal analysis and guarantee of generating edge cases. Thirdly, our method assumes access to a distribution of environments. Obtaining such realistic distributions [37, 38, 29] for simulation remain an open problem for the community.

**Acknowledgments**

We thank Dylan Hadfield-Menell, Ellis Ratner and Jessy Lin for their feedbacks on this work. This project is supported by a National Science Foundation NRI, Air Force Office of Scientific Research (AFOSR), the Office of Naval Research (ONR-YIP).

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

# A Algorithm

**Algorithm.** We now present our main algorithm for Assisted Reward Design. At every iteration $i$, we use the Maximal Information acquisition functions to select the next environment $M_{i+1}$ and query the reward designer. Note that the environment space $\mathcal{M}_{\text{devel}}$ is continuous and high dimensional and it is intractable to exhaustively search through it. We thus use *uniform sampling* to select a candidate set $\mathcal{M}_{\text{cand}} \in \mathcal{M}_{\text{devel}}$. Note that one can employ different heuristics for selecting such candidate sets, and we leave the heuristic design to future work.

**Representing Belief Distribution.** We use particles to represent $\tilde{P}_i(w = w^*)$: at each step, we sample $N_p$ particles from $\tilde{P}_i(w = w^*)$, compute importance weights based on Eq. 1 for each particle, and resample using the importance weights.

---

**Algorithm 1** Assisted Reward Design via Info-Gathering

---

Require prior $P_0(w)$, $\mathcal{M}_{\text{devel}}$, $N_{\text{cand}}$, $N_p$, initial training environments $\mathcal{M}_0$
Initialize posterior $\tilde{P}_0(w = w^*) = P_0(w)$
**for** $i = 0, .., T$ **do**
    $\tilde{w}_i \sim P_{\text{user}}(w|w^*, \mathcal{M}_i)$ { Query the designer on $\mathcal{M}_i$ }
    Compute posterior $\tilde{P}_{i+1}(w = w^*)$ using Eq. 4 or Eq. 5
    Sample $N_{\text{cand}}$ candidate environments $\mathcal{M}_{\text{cand}} \subseteq \mathcal{M}_{\text{devel}}$
    **for** $M \in \mathcal{M}_{\text{cand}}$ **do**
        Compute $f(M)$
    **end**
    Select $M_{i+1} = \arg\max_{M \in \mathcal{M}_{\text{cand}}} f(M)$
    $\mathcal{M}_{i+1} = \mathcal{M}_i \cup \{M_{i+1}\}$
**end**

---

**Complexity.** Our algorithm is bounded by the number of particles $N_p$ for representing posterior distribution and the number of candidates $N_{\text{cand}}$. At every iteration, the algorithm needs to solve for each particle $w$ in each candidate environment, which leads to a total of $O(N_p \cdot N_{\text{cand}})$ planning problems in order to compute belief update. This is the main speed limit. While one can potentially speed up by learning fast planners [39], we leave this to future work. In our experiments, we implement the environment as in Sec. B such that the dynamics and reward function are both vectorizable. We then concatenate the reward functions of $O(N_p \cdot N_{\text{cand}})$ problems and compute batch forward planning using gradient-based planner. Note that this is not feasible for general planning problems with non-differentiable dynamics or reward functions.

**Adding New Features.** When the designer adds in new feature $\phi_{k+1}$ on environment $M_n$, the new proxy reward has $d + 1$ dimensions while the previous proxies have $d$ dimensions. We can still perform reward inference on all of the $d + 1$ dimensions, as long as we incorporate all proxies received so far. The meta-agent would then have to revisit all the reward designs it's gotten so far and recompute posterior over the new augmented space. To do so, we invert Eq. 1 using the formula from [1]:

$$P(w = w^* | \tilde{w}_{1:n}, \tilde{M}_{1:n}) \propto P(w) \prod_{i=1:n} \frac{\exp(\beta w^T \tilde{\phi}_i)}{\tilde{Z}}, \tilde{Z}(w) = \int_{\hat{w}} \exp(\beta w^T \hat{\phi}_i) d\hat{w}$$

with $P(w)$ being the prior. We use MCMC to infer the distribution. During inference, every sample $w$ is of $d + 1$ dimensions, and $\tilde{\phi}_i$ are of $d + 1$ dimensions, computed from all previous proxy rewards (of $i$ or $i + 1$ dimensions) on the existing tasks. To compute the normalizer, we integrate over $\hat{w}$ in the $d + 1$ dimensional space.

# B Driving Environment Implementation

This section provides the details of the driving environment used in the paper. We introduce definitions of environment dynamics, feature and reward functions, the environment distributions, and how we implement the environment efficiently for trajectory optimization.

| Environment Features | | | |
|---|---|---|---|
| Feature Name | Raw Feature $\phi_{\text{raw}}$ | Transformation $\phi_{\text{full}}$ | Meaning |
| Speed | $v$ | $-(\phi_{\text{raw}} - v_{\text{goal}})^2$ | How much the vehicle deviates from the goal speed. |
| Control | $[u_{\text{steer}}, u_{\text{acc}}]$ | $-\|\phi_{\text{raw}}\|^2$ | The control effort by the vehicle. |
| Lane | $x$ | $\mathcal{N}(\phi_{\text{raw}} - \vec{x}_{\text{goal}}, d_{\text{lane}})$ | How much the vehicle deviates from target lane center. |
| Car | $[x, y]$ | $-\sum_i \mathcal{N}(\phi_{\text{raw}} - \vec{x}_{\text{car i}}, d_{\text{car i}})$ | How much the vehicle is driving close to other vehicles. |
| Obstacle | $[x, y]$ | $-\sum_i \mathcal{N}(\phi_{\text{raw}} - \vec{x}_{\text{obs i}}, d_{\text{obs i}})$ | How much the vehicle is driving close to the obstacles. |
| Fence | $x$ | $-[x_{\text{fence left}} - \phi_{\text{raw}}]_+$ $-[\phi_{\text{raw}} - x_{\text{fence right}}]_+$ | How much the vechile is outside the fence |

Table 1: Environment Feature Table

**Dynamics** We represent the forward dynamics of the MDP as $x_{t+1} = f(x_t, u_t)$, where we have the full knowledge of $x$. To model system dynamics, we use simple point-mass vehicle model with holonomic constraint. Let $x_{\text{car}} = [x, y, \theta, v]^\top$, where $x, y$ are the corrdinates of the vehicle, $\theta$ is the heading, and $v$ is the speed. We let $u = [u_{\text{steer}}, u_{\text{acc}}]^\top$ be the control input, where $u_{\text{steer}}$ is the steering input and $u_{\text{acc}}$ is the acceleration. We also use $\alpha$ as the friction coefficient. The model for a single vehicle is:

$$[\dot{x} \quad \dot{y} \quad \dot{\theta} \quad \dot{v}] = [v \cdot \cos(\theta) \quad v \cdot \sin(\theta) \quad u_{\text{steer}} \quad u_{\text{acc}} - \alpha \cdot v] \tag{6}$$

We model human vehicles as moving forward at constant speed and traffic cones as static objects.

**Feature and Reward Functions.** In the following Table 1 we introduce the environment features $\phi$ in the driving environment. Each feature composes of $\phi_{\text{raw}}$, a subset of the current state $x_t$ and control $u_t$, and a non-linear transformation on $\phi_{\text{raw}}$. The nonlinear transformation is designed such that when maximized, it induces the desired behaviour in that feature, such as moving to the target lane. These environment features are are differentiable and that we can characterize the desired driving behavior via a linear weighed sum $w^\top \phi$.

**Violations.** Here we provide in Table 2 the definition of environment constraint functions used in the experiment section to evaluate driving quality. Each constraint is a boolean function that returns true or false for one timestep, and we compute the final violation count for each trajectory by summing over constraints over the time horizon. These constraint functions are not used for optimization, but as an evaluation criterion for the case study experiment. Lower violation counts correspond to better driving behavior.

| Environment Constraints | | |
|---|---|---|
| Feature Name | Definition | Meaning |
| Overspeed | $v > v_{\text{max}}$ | The vehicle is driving over the maximal allowed speed. |
| Underspeed | $v < v_{\text{min}}$ | The vehicle is driving below the minimum speed on highway (i.e. backing up). |
| Uncomfortable | $\|u\|_\infty > \|u_{\text{max}}\|_\infty$ | The vehicle is applying too much force that it's uncomfortable (i.e. accerlating too much or jerky). |
| Collision | $\|\vec{x} - \vec{x}_{\text{car i}}\| \leq d_{\text{min}}$ | The vehicle crashes into the other vehicles. |
| Crash Object | $\|\vec{x} - \vec{x}_{\text{obj i}}\| \leq d_{\text{min}}$ | The vehicle crashes into the obstacles. |
| Offtrack | $x < x_{\text{fence left}}$ or $x > x_{\text{fence right}}$ | The vehicle drives off the left or the right fence. |
| Wronglane | $\|x - x_{\text{lane left}}\| \leq d_{\text{lane}}$ | The vehicle drives drives onto the wrong lane while merging. |

Table 2: Environment Constraints Table

**Planning Speed.** We implement the dynamics and reward function using JAX [40] and leverage the JIT compilation to speed up running time. We use shooting method and perform gradient-based planning using the Adam optimizer for 200 steps. We plan at a horizon of 10 timesteps and replan

every 5 timesteps. When planning for a single environment, we can generate 2.57 trajectories per second. Because we can vectorize the planning problem and plan for multiple trajectories at once, we can achieve 157.72 trajectories per second, by planning for 500 trajectories in batch. Computations are benchmarked on the c2-standard-4 instance on Google Cloud. We use batch planning for computing the belief distribution and environment proposals, as discussed in Sec. A as the main bottleneck of our algorithm.

**Environment Distribution.** We use a simple method to define the distribution of environments $\mathcal{M}_{\text{design}}, \mathcal{M}_{\text{deploy}}$. There are two human vehicles and two traffic cones positioned on the three-lane highway. We assume that the autonomous vehicle starts from the center of the scene ($x = 0, y = 0$), and sample the starting position of the other vehicles and obstacles. The other vehicles can start anywhere in $x_{\text{min car}} \leq x_{\text{car}} \leq x_{\text{max car}}, y_{\text{min car}} \leq y \leq y_{\text{max car}}$ and the obstacles can be initialized anywhere in can start anywhere in $x_{\text{min obs}} \leq x_{\text{obs}} \leq x_{\text{max obs}}, y_{\text{min obs}} \leq y \leq y_{\text{max obs}}$. We filter out situations where the other vehicles or obstacles are initialized to be colliding with the main vehicle.

Note that this is a very coarsely designed distribution to showcase our method. We have several limitations. For instance, we do not exclude the environments where the other vehicles runs into obstacles. These are relatively unlikely environments, and in principle we can define more realistic distributions by more careful environment engineering or by loading real world driving datasets.

The difference between $\mathcal{M}_{\text{devel}}$ and $\mathcal{M}_{\text{deploy}}$ is that in $\mathcal{M}_{\text{deploy}}$, we define a tighter feasible range of $x_{\text{car}}$ centered around the autonomous vehicle. This results in a shifted distribution of pseudo-difficulty metric with long-tail events. After discretization, $\mathcal{M}_{\text{devel}}$ has 274k environments and $\mathcal{M}_{\text{test}}$ has 91k environments. Given these large number of possible environments, it is impossible for reward designers to enumerate them manually.

## C    Experiment Details

**Optimal Planning** For the autonomous driving task, we compute trajectories of 10 timesteps. We use finite-horizon optimal planning with regard to given rewards. We plan at a horizon of 10 timesteps, and replan every 5 timesteps.

**Reward Design** We use rationality factor $\beta = \frac{0.1}{||\mathcal{M}||}$ to simulate noisy designer, and $\beta = 1$ in the inverse model. This is because we find empirically that the proxy reward quickly approaches the ground truth reward when we have multiple proxy environments. We thus divide the rationality factor $\beta$ it by the number of proxy environments to maintain its noisiness. To compute the posterior probability in Eq. 4 or Eq. 5, we need to approximate the normalizing constant. We follow the approach in [1] by sampling $w$. In the section 4.2 of [1], they find empirically that it helped to include the candidate sample $w$ in the normalizing sum. This requires planning with $w$ and computing its feature sum in the MCMC inner loop, which largely slows down the inference. Instead, we include the candidate $w$, but multiplies it with proxy $\tilde{w}$'s feature sum in the normalizing constant.

We then compare three methods (random, difficulty, maximal information) in iterative fashions. We use the same initial environment for the three methods and perform 9 iterations. We aggregate results over five random seeds.

**Environment Proposal.** During environment proposal, we use $N_p = 500$ particles on $N_{\text{cand}} = 64$ environments. We implement a vectorized car dynamic model using JAX [40] and Ray [41] for batch planning. Our environment proposal step takes 4 minutes on c2-standard-4 instance on Google Cloud.

**Evaluation.** To examine the quality of our inferred posterior of the reward function, we need to evaluate the posterior in new driving environments. We thus sample 500 environments uniformly from $\mathcal{M}_{\text{deploy}}$ as the deployment set for evaluation. We compute the regret, note that because of the different placement of vehicles and objects, different environment have different maximum and minimum reward they can produce. Thus it is unfair to compute the absolute regret $r_{\text{max}} - r_w$. We thus compute the relative regret, by first taking the worst cases reward $r_{\text{min}}$ in each environment. Then the relative regret is $\text{regret}_w = (r_{\text{max}} - r_w)/(r_{\text{max}} - r_{\text{min}}) = (r_{w^*} - r_w)/(r_{w^*} - r_{\text{min}})$.

