# OpenReview forum: "Assisted Robust Reward Design"
_robot-learning.org/CoRL/2021/Conference — CoRL2021 Poster_

### Official Review · Reviewer_J4qS · 2021-07-22

**Originality:** Good
**Technical Quality:** Very Good
**Clarity Of Presentation:** Excellent
**Impact:** 4

**Recommendation:**

Strong Accept: I recommend accepting the paper and will argue for my recommendation even if other reviewers hold a different opinion.

**Summary:**

This paper studies the problem of reward design for tasks where there is no inherent notion of the correct reward. The expert/user is expected to provide the reward function as weighted combination of pre-defined features such that it would result in the learner following desired trajectory. The learner maintains an epistemic uncertainty about the true reward function and models it as a belief over candidate weights. The learner then proposes a new environment to elicit a new input from the user based on the principle of maximum information gain from a predefined distribution (or collection) of domains, and this process continues iteratively. The primary claims of the authors are as follows:

1. An active learning approach leads to better identification of `misbehaving' environments than just considering the most difficult environments from the development distribution, or by selecting domains at random. This is borne out very well from simulation experiments and user studies.


**Issues:**

- I would request the authors to better distinguish their work from prior work in active reward learning. I understand that the work is novel, but given there is significant research out there on active learning for rewards and more generally for task specifications, the novelty of the work should be made more explicit.

- I would also urge the authors to include characterization of the problem domain distribution, and efficient inference over it as yet another open challenge

**Reviewer Expertise:**

Very good: Comprehensive knowledge of the area

**Strengths And Weaknesses:**

**Strengths :**

- Reward design is an important yet under-addressed issue for RL: Reward functions for many real world tasks are seldom set in stone, and are notoriously hard to formalize. There is a strong need to consider approaches that reason about the epistemic uncertainty of the true task specifications (in this case Markov rewards, but the issue extends to many task specification formalisms). This work tackles this important problem, and has promising results albeit in a toy domain.

- Active Learning Formulation through information gain heuristic: The paper provides strong evidence that information gain is a useful heuristic for active learning within Markov reward domains. It complements prior research evidence that indicate towards information gain being a useful heuristic. This paper instantiates an info-gain based active learning approach to direct reward elicitation that to my knowledge is novel.

- Intuitive results: The active learning approach identifies domains that actively try to 'break' the current reward function. This is what an active learner would be expected to perform.

**Methodological Comments:**

- It seems that the burden of designer shifts from specifying the right reward distribution to capturing the right distribution over domains. It appears that in this work, the authors assume access to a dataset of domains, and sample from that dataset. But for domains where data over environments is sparse, this approach would require a generative model over the domains. It would be nice to see an example where the problem distribution is described as a generative model.

- The authors have used particle filtering based techniques, and seem to assume a unimodal distribution over users reward function feedback model. I wonder if that would be true in all cases. If the `true' reward function is non-unique, would the particle filtering approach still recover all the modes? Would the potentially conficting feedback throw the learner off?

- This method relies on active learning queries that necessarily result in failure during deployment. Such a technique would me most amenable to domains where simulators are readily available. However active learning can also be valuable in domains where lack of simulators mean new policies need to be learnt on the fly during deployment. Can the info-gain heuristic be modified to generate executions that result in low performance  but not outright failure? (Note this is not a weakness, just a question to the authors)

- There are many heuristics for active learning (uncertainty sampling, max model change, variance reduction, query-by-committee etc.) [1], have the authors considered other heuristics, but selected information gain? If not it would be interesting future work to explore the alignment of these heuristics within the Markov reward domain considered by the authors.

[1] - Settles, Burr. "Active learning literature survey." (2009).

**Summary Of Recommendation:**

I would recommend the acceptance of this paper. This is an under-explored problem, and to my knowledge a novel approach to the problem of reward design.

---

### Official Review · Reviewer_5qJa · 2021-07-23

**Originality:** Very Good
**Technical Quality:** Very Good
**Clarity Of Presentation:** Very Good
**Impact:** 4

**Recommendation:**

Weak Accept: I recommend accepting the paper, but will not argue for my recommendation if the majority of other reviewers have a different opinion.

**Summary:**

This paper formalizes the reward design problem for robotic tasks as an iterative process. A human engineer can specify a reward signal for the robot, and the agent can propose environments that highlight potential unexpected consequences of optimizing that reward signal, enabling the human engineer to iteratively change the reward function until the desired behavior is consistent across environments. This paper also proposes an approximate algorithm for tractably estimating the reward function based on a particle-filter approach for representing uncertainty about the true reward function.

**Issues:**

The authors should fix the references section so that it is consistent and accurate with what publications were cited.

**Reviewer Expertise:**

Fair: Some knowledge of the area

**Strengths And Weaknesses:**

Strengths
- Treating the reward design process as an explicit problem that the robot agent can help guide in is very interesting, and addresses the difficult problem of creating reward signals for autonomous robots in complex environments. This is a very interesting paper and idea to tackle.
- The assisted reward design problem is well-formulated and easy to follow, and the observation model for incorporating the human engineers reward function into what the robot thinks the actual reward function is principled and based on previous work in Inverse Reward Design.
- The approximate strategy for proposing environments that provide the most information gain over the belief distribution over reward functions is interesting and is a good solution to the formulated problem.

Weakness:
- In the related work, the authors claim: "Note that in many real-world problems, such as autonomous cars, there are dangerous situations where no ground-truth rewards exist". The authors should clarify this statement, since it's not clear if this implies that the authors believe RL is not applicable to these kind of tasks.
- The references should be addressed and properly presented (certain works are missing years [22],[25] and many are missing where they were published)


**Summary Of Recommendation:**

This paper presents an interesting and useful formulation of reward specification as an iterative process and proposes a tractable method for aiding users through the reward design process. This paper is easy to read and presents an interesting approach, and I recommend it be accepted.

---

### Official Review · Reviewer_oemY · 2021-07-24

**Originality:** Very Good
**Technical Quality:** Very Good
**Clarity Of Presentation:** Good
**Impact:** 4

**Recommendation:**

Weak Accept: I recommend accepting the paper, but will not argue for my recommendation if the majority of other reviewers have a different opinion.

**Summary:**

This paper formulates reward design as an iterative procedure between a learning agent and a designer. It assumes the existence of a ground-truth reward function for a distribution of environments. The agent treats the reward definition from the designer as an evidence and tries to form a belief over the ground-truth reward. Solution: at any given iteration, the agent proposes an environment to incentivize the designer to specify a reward which provides maximal information about the true reward. For example, in autonomous driving, the agent could present a scenario where the agent trajectory brings it dangerously close to other cars. The designer would have to carefully craft a reward function, that helps the agent quickly improve its belief over the ground-truth reward. Experiments with simulated and human designers on a toy environment show that assisted reward design gets closer to ground-truth reward when compared against baselines which propose the most difficult environments or random environments.

**Issues:**

1. I hope the authors can justify the assumptions of the problem setup and respond to their limitations.
2. The authors can add more details to the writing to make it more approachable for the reader.
3. Some explanation on how to extend this setup to realistic scenarios is required, especially what are the next immediate environments to consider?

Details for each of these issues are listed under weaknesses above.

**Reviewer Expertise:**

Good: General knowledge of the area

**Strengths And Weaknesses:**

# Strengths
1. **Novelty**: The assisted reward design setup is novel, to the best of my knowledge. It is a meaningful problem because reward design is indeed an iterative procedure in practice. Thus, it makes sense to assist a designer by actively querying rewards for information maximizing scenarios.
2. **Human Evaluation**: The experiments on simulated and human designers are positive and consistent. This validates the problem setup and method well.
3. **Qualitative Results**: I appreciate the analyses and examples explored by the paper to demonstrate convincingly how maximal information gain is helping the reward design to be efficient.

# Weaknesses
1. **Major assumptions**

The paper makes major assumptions in the problem setup, which are usually unrealistic.

- *Environment*: A known distribution of environments is assumed which can be sampled from repeatedly. Thus, the setup does not work for a single environment/MDP, which is most often the case in reinforcement learning.
Even if we wish to somehow convert a single environment to a distribution,
this is only possible in simulation, and only when when the environment configurations can be parameterized.
The paper mentions large datasets to be valid as well, but (a) it is not feasible to relabel the entire dataset every time the designer changes the reward function, and (b) it is not possible to find the optimal trajectory from a static dataset without active environment interactions, when a new reward function is given.
- *Ground-Truth Reward*: The paper assumes that there is a ground-truth reward function governing the true behavior on the entire distribution of environments. This usually does not hold. For instance, if I am training an agent to walk on diverse terrains, there is no concept of a ground-truth reward. A reward specifying agent stability and distance covered could be too unspecified that it doesn't help training, and a reward too detailed would become specific to each terrain so it is no longer common for all environments. This brings me to the next assumption...
- *Parameterized Reward*: The reward function should be defined on pre-defined features and the designer's job is limited to changing the weights. This is limiting because most iterative reward engineering (in my experience) involves figuring out some edge cases or special scenarios. The fix usually requires extra dimensions to be added to the reward or defining conditional statements.
- *World Model & Planning algorithms*: The paper acknowledges that access to perfect world models and efficient planning algorithms can be challenging. I just want to reiterate that training RL agents on every particle sample representing a reward on a large variety of environments for every iteration will blow up in complexity.


2. **Writing**: The paper is well-written as a whole, but there are parts in the method section which become hard to follow. I understand it is difficult to compress all the information in 8 pages, but still it would assist the reader if more details about the method are present in the paper. For example:
- intuition behind Eq. (1).
- how eq. (4) and (5) are used to update beliefs.
- moving the algorithm from appendix to the main paper.
- where and how to use world models and RL policy in the method.
- in figure 1, the captions or figure should describe what w, w~ and M represent.


3. **Evaluation**: The paper only provides results on a proof-of-concept toy environment. It would be nice to see how assisted reward design scales to more complex control problems to really demonstrate how this can be a valuable procedure to the community in the near-future. I do not consider this a major weakness, since the concept seems novel enough to me. But, could the authors at least suggest some other environments where this proposed setup is already feasible to use, so we can imagine a bridge to realistic problems?

**Summary Of Recommendation:**

Despite the assumptions and toy-evaluation, I believe the concept presented in this paper is novel and potentially valuable in the future. The paper includes human designer evaluation which aptly shows the correctness and benefits of this method. After a few improvements to writing, I recommend this paper to be accepted on these grounds.

===== Response After Rebuttal (No Change) =====
Thank you for your response. I am satisfied with it and keep my recommendation of weak accept. Please accommodate any leftover writing and clarity changes as pointed by other reviewers.

---

### Official Review · Reviewer_pnb1 · 2021-07-24

**Originality:** Good
**Technical Quality:** Fair
**Clarity Of Presentation:** Fair
**Impact:** 1

**Recommendation:**

Weak Reject: I recommend rejecting the paper, but will not argue for my recommendation if the majority of other reviewers have a different opinion.

**Summary:**

The paper presents assisted reward design method where the reward is a weighted function of different components and the goal is to find the best reward function that is optimal for all different possible scenarios. The context is similar to active learning, where the user provides parameters for the reward function, while the algorithm tries to find set of scenarios that is provided to the user. This is an iterative process and the algorithm tries to make the user find the reward function that is optimal for all possible scenarios (even the ones not provided to the user by the algorithm). The authors show their results on an autonomous driving simulation where the car is trying to switch to the left lane with existence of different cars and obstacles. The results show that experimenting with 12 experts, the algorithm finds edge cases for different

**Issues:**

I think the main issues are the terminology, the fit to the robotics domains and the evaluation of the method. Please see the comments above  for details.

**Reviewer Expertise:**

Very good: Comprehensive knowledge of the area

**Strengths And Weaknesses:**

The paper is well written and it does a good job at covering basics and relevant publications. The main strength of the paper is that they take a different approach to the reward design.
At times, I found the paper to be hard to understand due to the terms used. Maybe it is because of my background with reinforcement learning, but the paper usesthe terms "reward design", "state", "action" for:
 - cost function that is used to calculate the behavior
 - set of environments
 - adding environment to the set
which makes it very confusing as these terms are commonly used in reinforcement learning, but here there is no learning of a policy in the defined setting. Also, the authors use the term "robot" to define an agent that selects the set of environments,which is very uncommon (also not sure why the authors preferred the term robot for it). In my opinion the paper would be significantly more clear if the users drop the terms "robot", "state", "reward design". If I understand the concepts correctly, it presents an algorithm that selects set of scenarios such that if a person would optimize weights of a cost function based on these scenarios, behaviors based on that cost function will perform good in the set of environments defined by parameters.

The selected environment (autonomous driving) contains a single scenario of "autonomous driving" by looking at moving to the left lane. It provides unlimited amount of scenarios due to environment parameters such as obstacle or other cars. The other cars are driven with fixed speed, and they don't react to the car that changes lane using this cost function. Considering this, I think that presenting the domain as "autonomous driving" is an overstatement.

The experimentation is done with "11 robotics expert" to design the parameters of the reward function. I was wondering if being a robotics expert is needed, or any regular person who drives a car maybe could use this framework to optimize the cost function. This is mainly because the goal of the algorithm presented in this paper is to assist the user in designing the weights.

Also, I was wondering if any blackbox algorithm or optimization could be used to optimize these weights instead of using a robotics expert. This would allow, thousands or millions of iterations between the algorithm and the designer, and would be fully automated.

The reward design problem is actually very interesting within the reinforcement learning context. If the authors change the problem so that the agent (or robot) is the car, and it behaves based on some observations (i.e. lidar or features), we would use reinforcement learning to train the car. In this case, the problem of reward design would become "reinforcement learning while also finding the optimal reward function" and be comparable to other works in the literature such as intelligent environment design or learning to learn. This would also make the problem more applicable to other real world domains.

**Summary Of Recommendation:**

The paper touches an interesting problem of designing the reward function with human in the loop. On the other hand, the framing of the terms are very confusing (i.e. robot, state, action). The authors define environment selection algorithm as the robot, but I do not think that this definition would be applicable to problems in robotics. The tested domain is a very simple scenario (switching to lane), despite its potential difficulty due to unlimited number of scenarios with different parameters, I don't think it is representative of the problem of autonomous driving.
The paper has an interesting direction and good potential in the context of active learning, but I don't think that it fits the robotics framework that the authors try to use. I would also expect it to be evaluated on more complex problems where learning might be needed using the optimized / designed reward function.

---

### Meta-Review · Area_Chair_kyMC · 2021-08-11

**Recommendation:** Accept (Poster)
**Confidence:** 4

**Metareview:**

This paper proposes an actively learning approach for iterative reward design in reinforcement learning. All reviewers acknowledge that the problem is interesting and under-explored. This paper has presented novel solutions to this important problem, makes good technical contributions, and has promising results albeit in a toy domain. After the revision, reviewers' original concerns about use-of-terminologies, and strong assumptions were addressed. Please accommodate remaining writing and clarity changes, and explicit statements contrasting with prior work, as pointed by reviewers, in the camera-ready version of the paper.

---

### Decision · Program_Chairs · 2021-09-13

**Decision:**

Accept (Poster)

**Comment:**

This paper proposes an actively learning approach for iterative reward design in reinforcement learning. All reviewers acknowledge that the problem is interesting and under-explored. This paper has presented novel solutions to this important problem, makes good technical contributions, and has promising results albeit in a toy domain. After the revision, reviewers' original concerns about use-of-terminologies, and strong assumptions were addressed. Please accommodate remaining writing and clarity changes, and explicit statements contrasting with prior work, as pointed by reviewers, in the camera-ready version of the paper.